# Evaluation of the Bone Regeneration Effect of Recombinant Human Bone Morphogenic Protein-2 on Subperiosteal Bone Graft in the Rat Calvarial Model

**DOI:** 10.3390/ma12101613

**Published:** 2019-05-16

**Authors:** Eunhye Jang, Ja-Youn Lee, Eun-Young Lee, Hyun Seok

**Affiliations:** 1Department of Orthodontics, College of Dentistry, Gangneung-Wonju National University, Gangneung 25457, Korea; jianejiane@naver.com; 2Department of Prosthodontics, Chungbuk National University Hospital, Cheongju 28644, Korea; jayoun.lee@gmail.com; 3Department of Oral and Maxillofacial Surgery, Chungbuk National University College of Medicine, Cheongju 28644, Korea; ley926@chungbuk.ac.kr; 4Department of Oral and Maxillofacial Surgery, Chungbuk National University Hospital, Cheongju 28644, Korea

**Keywords:** bone morphogenetic protein, bone regeneration, subperiosteal bone graft, bone sialoprotein, osteocalcin

## Abstract

The aim of this study was to evaluate the bone regeneration effect of recombinant human bone morphogenetic protein-2 (rhBMP-2) on a subperiosteal bone graft in a rat model. A subperiosteal space was made on the rat calvarium, and anorganic bovine bone (ABB), ABB/low bone morphogenetic protein (BMP) (5 µg), and ABB/high BMP (50 µg) were grafted as subperiosteal bone grafts. The new bone formation parameters of bone volume (BV), bone mineral density (BMD), trabecular thickness (TbTh), and trabecular spacing (TbSp) were evaluated by microcomputed tomography (µ-CT), and a histomorphometric analysis was performed to evaluate the new bone formation area. The expression of osteogenic markers, such as bone sialoprotein (BSP) and osteocalcin, were evaluated by immunohistochemistry (IHC). The ABB/high BMP group showed significantly higher BV than the ABB/low BMP (*p* = 0.004) and control groups (*p* = 0.000) and higher TbTh than the control group (*p* = 0.000). The ABB/low BMP group showed significantly higher BV, BMD, and TbTh than the control group (*p* = 0.002, 0.042, and 0.000, respectively). The histomorphometry showed significantly higher bone formation in the ABB/low and high BMP groups than in the control group (*p* = 0.000). IHC showed a high expression of BSP and osteocalcin in the ABB/low and high BMP groups. Subperiosteal bone grafts with ABB and rhBMP-2 have not been studied. In our study, we confirmed that rhBMP-2 contributes to new bone formation in a subperiosteal bone graft with ABB.

## 1. Introduction

Bone graft operations in the oral and maxillofacial regions have been performed in bone defect areas where a cyst, tumor, impacted tooth, trauma, or cleft alveolus was present [1,2]. Bone grafts for dental implants have been made on the surface of the alveolar bone but not in the bone defect. Subperiosteal bone grafts on the surface of the alveolar cortical bone have been performed in the atrophic alveolar bone [3,4]. Sufficient alveolar bone height and width are required for the success of dental implants, and the implant surface should be surrounded by alveolar bone [5]. Vertical and horizontal bone augmentation is necessary in the alveolar bone to obtain an adequate bone volume for implant installation.

An autogenous bone is an ideal bone graft material for the reconstruction of a bone defect because of its bone regeneration abilities of osteogenesis, osteoinduction, and osteoconduction [6,7]. It has resistance to bacterial infection, with no immune response on the recipient site. However, it has clinical limitations, such as the risk of donor site morbidity, limited harvesting bone volume, and a large amount of resorption on the recipient site [5]. For these reasons, a bone substitute material has been developed and is widely used in bone grafts for implant installation [8]. Anorganic bovine bone (ABB), as the commercial product of a xenograft, shows high osteoconductive ability and a low resorption rate [9]. It has a suitable mineral content that can be integrated with human bone and an optimal micropore structure to facilitate vascular ingrowth and new bone formation [10]. It has been widely used as a bone substitute for implant surgery; however, ABB only provides osteoconductive ability and not osteogenetic or osteoinductive abilities [8]. Therefore, the combined use of ABB with an osteoinductive growth factor, such as the bone morphogenetic protein (BMP), would be necessary for new bone formation.

BMPs, as an osteoinductive growth factor, induce the differentiation of the osteogenic progenitor cell into the mature osteoblast [11,12]. The osteoinductive ability of several kinds of BMPs, such as BMP-2, -4, and -7, has been reported [13]. Particularly, recombinant human bone morphogenetic protein-2 (rhBMP-2) has been widely used in bone graft operations for dental implants, such as sinus elevation, alveolar bone augmentation, and socket preservation, after tooth extraction [12,14,15,16]. Bone augmentation using only ABB is difficult to achieve due to a lack of osteoinductive ability. Therefore, rhBMP-2 can be used with an osteoconductive scaffold to the bone graft area to promote new bone regeneration.

Subperiosteal bone grafting is a minimally invasive technique that has been performed in oral surgery for new bone formation in the atrophic alveolar bone [17]. The bone graft operation on the subperiosteal space or pocket in the alveolus shows a favorable result in new bone formation, and it regenerates enough bone to install dental implants [18]. Various bone substitute materials are used in subperiosteal bone grafts in the alveolus; however, the combined use of a bone substitute and rhBMP-2 has not been evaluated. rhBMP-2 can be easily and simply used in subperiosteal bone grafts with a bone substitute. We used the subperiosteal space on the rat calvarial bone to evaluate the bone regeneration ability of the graft material [19]. In addition, ABB and rhBMP-2 were applied on the rat calvarial surface to evaluate the bone regeneration effect of rhBMP-2 in subperiosteal bone grafts.

The aim of this study was to evaluate the effect of rhBMP-2 in new bone regeneration on a subperiosteal bone graft in a rat calvarial model. We performed the bone graft on the subperiosteal space of the rat calvarium by using ABB alone and ABB soaked with low and high doses of rhBMP-2. For an evaluation of the bone regeneration ability of rhBMP-2, microcomputed tomography (µ-CT) was used to measure new bone formation, and a histomorphometric evaluation was performed to evaluate the newly formed bone area. In addition, immunohistochemistry (IHC) of bone regeneration markers was also performed.

## 2. Materials and Methods

### 2.1. Experimental Animals and Study Design

A total of 18 12-week-old Sprague Dawley rats (Samtako Biokorea, Osan, Korea) with average weights of 250 g (between 200 and 300 g) were used in this study. The rats were housed two per cage under specific pathogen-free conditions and were fed a standard rodent diet and water ad libitum. The animals were acclimated to the new environment for 14 days before the surgery. This study was approved by the Institutional Animal Care and Use Committees of Chungbuk National University, Cheongju, Korea (CBNUR-1109-18).

The rats were randomly divided into three groups (n = 6), and a subperiosteal pocket was created on the parietal bone of each rat calvarium. rhBMP-2 (COWELL^®^ BMP; Cowellmedi, Busan, Korea) and anorganic bovine bone (ABB; Bio-Oss^®^, Geistlich Pharma AG, Wolhusen, Switzerland) were used in this study. The rhBMP-2 powder was diluted with saline, and low (5 µg) and high (50 µg) doses of rhBMP-2 were prepared. In the ABB group (n = 6), ABB (0.08 g) alone was grafted onto the subperiosteal calvarial pocket for the control group, and ABB (0.08 g) with 5 µg and 50 µg of rhBMP-2 were grafted onto the same space as the ABB/low BMP (n = 6) and ABB/high BMP (n = 6) group.

### 2.2. Surgical Procedure

An intramuscular injection of a combination of zolazepam (15 mg/kg; Zoletil 50^®^, Vibac, Carros, France) and xylazine (0.2 mL/kg; Rumpun^®^, Bayer Korea, Seoul, Korea) was performed for general anesthesia. The cranial area of the skull was shaved and then disinfected with povidone-iodine. Local anesthesia was performed on the cranial area with an injection of 2% lidocaine with epinephrine (1:100,000). A horizontal step incision was made on the posterior portion of the calvarial bone. Sharp subperiosteal dissection was performed to reflect the pericranium and to expose the calvarial bone. A subperiosteal pocket was made on the parietal bone of the calvarium, and the ABB alone was grafted into the pocket as a control group (Figure 1). After the ABB was grafted, the diluted rhBMP-2 (5 and 50 µg) was applied onto the ABB material in the subperiosteal pocket. After grafting, the muscle and skin were closed with 3-0 Vicryl (ETHICON, Somerville, NJ, USA). Gentamicin (1 mg/kg; Kookje, Seoul, Korea) and sulpyrine (0.5 mL/kg; Green Cross Veterinary Products, Seoul, Korea) were injected intramuscularly three times daily for 3 days. Six rats from each group were sacrificed at 6 weeks after surgery. Specimens were fixed in 10% formalin. μ-CT and a histological analysis were performed for the evaluation of new bone formation.

### 2.3. μ-CT Analysis

All calvarial samples were analyzed by μ-CT at the Osong Medical Innovation Foundation Laboratory Animal Center (Osong-eup, Cheongju-si, Korea) and Ochang Center of the Korea Basic Science Institute (Ochang-eup, Cheongju-si, Korea). The samples were taken by a Quantum FX micro-CT (Perkin Elmer, Waltham, MA, USA). The CT scanner was set to 90 kV for the X-ray tube, a 160-μA current for the X-ray source, and 120 s of exposure time. The detector and the X-ray source were rotated 360° in 360 rotation steps. The number of calibration exposures was 30. System magnification was performed over 24 mm of the axial field of view (FOV) and over 24 mm of the transaxial FOV. The scanned images were reconstructed by CTAn (Bruker, Hamburg, Germany). The region of interest (ROI) of each sample was the grafted area on the surface of the cranium of the rat, and it was reconstructed in 3D images for the analysis of bone volume (BV), bone mineral density (BMD), trabecular thickness (TbTh), and trabecular spacing (TbSp).

### 2.4. Histologic and Histomorphometric Analysis

After the μ-CT analysis, the samples were decalcified in 5% nitric acid for 2 weeks and dehydrated in ethyl alcohol and xylene. The samples were separated through the midline sagittal suture and embedded in paraffin blocks. The paraffin blocks were sliced into sections that were then stained with hematoxylin and eosin. The sections that showed the sagittal image of the parietal bone and bone grafted area were selected for histologic analysis. Digital images of the selected sections were taken with a digital camera (DP-73; Olympus, Tokyo, Japan). The images were analyzed by Sigma Scan pro (SPSS, Chicago, IL, USA). The amount of newly formed bone was calculated as the percentage of the total region of the bone graft area.

### 2.5. Immunohistochemical Analysis of Osteogenic Marker

IHC was performed on histological sections to evaluate the expression of osteogenic markers, such as bone sialoprotein (BSP) and osteocalcin. Anti-BSP (GTX12155; GeneTex) and antiosteocalcin (sc-365797; Santa Cruz Biotechnology) antibodies were used as primary antibodies. Immunohistochemical staining was performed using a Dako REAL EnVision Detection System (Dako, Glostrup, Denmark) according to the manufacturer’s protocols. Counterstaining was conducted with Mayer’s hematoxylin (Sigma-Aldrich, St. Louis, MI, USA). Stained tissue slides were examined with an Olympus BX51 (Olympus, Tokyo, Japan) microscope.

### 2.6. Statistical Analysis

One-way analysis of variance (ANOVA) was used for comparisons of three or more independent groups. Bonferroni’s method was used for post hoc tests. Differences with *p*-values of less than 0.05 were considered to be significant.

## 3. Results

### 3.1. Results of μ-CT Analysis

The 3D reconstruction images in each group are presented in Figure 2. The ABB was grafted onto the surface of the parietal bone, and it was well accumulated on the subperiosteal pocket in all groups. In the control group, the grafted ABB particle was located on the parietal bone without new bone formation (Figure 2a). The ABB/low and high BMP groups showed new bone formation that covered the grafted ABB particle (Figure 2b). The ABB/high BMP group showed more favorable bone regeneration than other groups, and it had a more compact and denser new bone surface (Figure 2c).

The μ-CT analysis result of each group is presented in Figure 3. The BV was 68.69 ± 12.18 mm^3^ in the control group, 103.42 ± 13.32 mm^3^ in the ABB/low BMP group, and 134.88 ± 15.24 mm^3^ in the ABB/high BMP group at 6 weeks after operation (Figure 3a). The BV of the ABB/high BMP group was significantly higher than that of the control group (*p* = 0.000) and the ABB/low BMP group (*p* = 0.004). In addition, the BV of the ABB/low BMP group was significantly higher than that of the control group (*p* = 0.002). The BMDs of the control, ABB/low BMP, and ABB/high BMP groups were 781.98 ± 4.03, 794.68 ± 9.14, and 793.64 ± 9.40 mg/cc (Figure 3b). The BMD of the ABB/low BMP group was significantly higher than that of the control group (*p* = 0.042).

The TbTh of the control, ABB/low BMP, and ABB/high BMP groups were 0.10, 0.11, and 0.12 ± 0.01 mm (Figure 3c). The TbTh of the ABB/low BMP and ABB/high BMP groups were significantly higher than that of the control group (*p* = 0.000). The TbSp of the control, ABB/low BMP, and ABB/high BMP groups were 0.10 ± 0.02, 0.07 ± 0.02, and 0.08 ± 0.02 mm (Figure 3d). The average TbSp of the control group was greater than that of other groups, but there was no significant difference (*p* > 0.05).

### 3.2. Results of Histologic and Histomorphometric Analyses

The histologic and histomorphometric analyses of each group are presented in Table 1 and Figure 4. In the histologic analysis, more new bone formation was observed around the bone graft material area in the ABB/low and high BMP groups. The newly formed bone grew into a space between the graft materials, and osteoblast and osteocyte were well arranged in the new bone (Figure 3). More new blood vessel formations were observed inside the new bone, and the newly formed bone was more matured and mineralized than that of the control group.

The new bone formation areas at 6 weeks after the operation in the control, ABB/low BMP, and ABB/high BMP groups were 11.78% ± 3.10%, 31.03% ± 2.50%, and 36.06% ± 6.92%, respectively, and the difference was statistically significant (*p* < 0.000). In the post hoc test, the new bone formation of the ABB/low BMP and ABB/high BMP groups were significantly higher than that of the control group (*p* = 0.000). However, there was no significant difference between the ABB/low BMP and ABB/high BMP groups (*p* = 0.235).

### 3.3. Results of Immunohistochemical Evaluation of the Osteogenic Marker

The IHC showed the greater expression of BSP and osteocalcin in the ABB/low and high BMP groups than in the control group. The BSP was highly expressed in the new bone matrix of the ABB/low and high BMP groups, whereas that of the control group was weakly expressed (Figure 5). The osteoblast and osteocyte in the new bone matrix of the ABB/low and high BMP groups had high osteocalcin expression (Figure 5).

## 4. Discussion

Bone graft operations for dental implants have been performed on the cortical surface of alveolar bone [20]. The sufficient alveolar bone height and width should be sufficient for dental implant installation [21]. The subperiosteal bone graft has been clinically performed as a subperiosteal tunneling technique in oral and maxillofacial surgery [18]. The subperiosteal tunneling technique involves making a subperiosteal space or pocket in the alveolar ridge via a small incision and dissection, and the bone substitute material is grafted under the subperiosteal space [22]. This technique is a minimally invasive bone graft procedure rather than a conventional bone graft operation [17]. The conventional bone graft approach is an incision of full thickness of oral gingiva, which reflects the full mucoperiosteal flap and accomplishes a bone graft with a direct view of the deficient alveolar bone. The subperiosteal bone graft has the advantage of fewer postoperative complications, including bleeding, infection, and bone graft failure, compared with conventional bone grafting [17]. Within the subperiosteal pocket, the grafted material can receive rich osteogenic cells and blood supply from the periosteum with the protection of surrounding soft tissue [23,24,25].

The bone substitute material has been grafted in the subperiosteal space in the atrophic alveolar bone. The combined use of the rhBMP-2 and bone substitute has not been studied. In this study, we grafted the ABB and rhBMP-2 in the rat calvarial surface via the subperiosteal space and evaluated the bone regeneration ability of rhBMP-2.

In the result of the μ-CT analysis, the ABB/low and high BMP groups showed more effective bone regeneration ability than did the control group (Figure 3). In addition, the dose of rhBMP-2 was significantly related with the amount of new bone formation. The total BV in the ABB/high BMP group was 134.88 ± 15.24 mm^3^, which was significantly higher than that of the ABB/low BMP group (103.42 ± 13.32 mm^3^, *p* = 0.004) and the control group (68.69 ± 12.18 mm^3^, *p* = 0.000). Furthermore, the BV of the ABB/low BMP group was also significantly higher than that of the control group (*p* = 0.002, Figure 3a). The other bone parameters showed that the use of rhBMP-2 had a positive effect on bone regeneration compared with only the use of ABB. The average BMD of the ABB/low and high groups was higher than that of the control group. In addition, there was a significant difference between the ABB/low BMP and control groups (Figure 3B). Thicker trabecular bone and smaller trabecular bone spacings were observed in the newly formed bone in the ABB/low and high BMP groups than that in the control group (Figure 3c,d). The bony ingrowth by the use of rhBMP-2 induced trabecular bone thickening and reduced bone spacing. The results of the μ-CT analysis mean that the use of rhBMP-2 in the subperiosteal bone graft had a positive effect on new bone regeneration compared with only the use of ABB. Similarly, with the result of μ-CT, the 3D reconstruction image of the ABB/high BMP group showed more dense and compact bone formation compared with that of the other groups (Figure 2).

BMPs are the superfamily protein of the transforming growth factor-β (TGF-β) and have an important role in the maintenance of adult tissue homeostasis [26,27]. BMPs have crucial roles in bone and cartilage formation, and BMP-2, -4, -6, and -7 have been known to have osteoinductive ability [28]. BMP-2 has strong osteoinductive potential and has been used as an osteoinductive growth factor [5]. In advance of genetic engineering, massive production of rhBMP-2 is possible by genetic recombination using *Escherichia coli* or mammalian cells [29]. rhBMP-2 stimulates the differentiation of the osteoblast from the mesenchymal cell, contributes to osteogenesis, and is now widely used in clinical applications [30]. Various bone substitute materials have been used as the rhBMP-2 carrier and scaffold. rhBMP-2 shows excellent osteoinductive ability and contributes to new bone formation in the bone defect combined with collagen sponge, hydroxyapatite, calcium phosphate, and bovine bone [5,11,31,32,33]. In our histological analysis, more new bone formation was observed in the ABB/low and high BMP groups than in the control group. In the ABB/high BMP group, the newly formed bone surrounded the bone graft material and the subperiosteal pocket (Figure 4). The ABB/high BMP group had more new bone ingrowth between the ABB particles and showed excellent bone maturation and mineralization. On the other hand, the control group had less bony ingrowth and reduced bone maturation. From this result, rhBMP-2 showed osteoinductive ability and contributed to the new bony ingrowth in the ABB particle. In addition, in the histomorphological analysis, there was a significant difference in the newly formed bone area among the three groups (*p* = 0.000). The total new bone area in the ABB/high BMP group was 36.06% ± 6.92%, which was significantly higher than that of the ABB/low BMP group (31.03% ± 2.50%) and the control group (11.78% ± 3.10%).

In the IHC, the expression of BSP and osteocalcin in the ABB/low and high BMP groups was greater than that of the control group. BSP is a bone extracellular matrix protein that is found in mineralized tissue, such as bone, dentin, and calcified cartilage [34]. BSP plays a crucial role in the intramembranous ossification and biomineralization of tissue [35]. The high expression of BSP was observed in the newly formed bone matrix of the ABB/low and high BMP groups, whereas the control group was weakly immunostained to the BSP (Figure 5b,c). Osteocalcin is a noncollagenous protein that is secreted in the osteoblast, is used as an osteoblast differentiation marker, and is characterized by osteogenesis [36]. The osteoblast and osteocyte in the new bone matrix in the ABB/low and high BMP groups were highly expressed, whereas the control group was weakly stained (Figure 5e,f). The high expression of osteocalcin in the ABB/low and high BMP groups showed the effect of rhBMP-2 in osteoblast differentiation. IHC confirmed that the rhBMP-2 effectively induced osteoblast differentiation, osteogenesis, new bone maturation, and mineralization combined with ABB in the subperiosteal bone graft.

The subperiosteal bone graft technique is a minimally invasive bone graft procedure that uses the subperiosteal space [22]. The subperiosteal bone graft has been performed on the atrophic alveolar bone ridge for augmentation of the vertical or horizontal length by using various bone substitutes, such as bovine bone, hydroxyapatite, and allograft material [17,18,22]. The subperiosteal space is surrounded by the periosteum, which has a rich mesenchymal stem cell, progenitor cell, and cytokines to induce bone formation [19,37]. The periosteal-derived cell has excellent osteogenic activity and is capable of osteoblast differentiation [38,39]. Thus, the preservation of the periosteum and bone graft under the periosteum is an effective method for new bone formation. rhBMP-2 has a water-soluble property and is dissolved in saline. rhBMP-2-soaked hydroxyapatite has been grafted onto the atrophic alveolar ridge for bone augmentation [5,40]. When the rhBMP-2 solution is applied to the alveolus via the conventional bone graft approach, which has a full gingival flap elevation, there is a risk of discharge of the rhBMP-2 solution during operation [5,40]. The early discharge of rhBMP-2 does not induce osteoinduction on the bone graft site and leads to failure of new bone formation. The subperiosteal bone graft preserves the periosteum, uses the small incision for an entrance, and dissects the pocket to make the bone graft space. In the subperiosteal space, rhBMP-2 and the bone substitute can be better contained and stabilized without discharge, and it could more effectively induce new bone formation compared with that of the conventional bone graft approach.

As we expected, the group that used the high dose of rhBMP-2 showed greater new bone formation than other groups. The ABB/high BMP group showed higher BV and TbTh in the μ-CT analysis, more new bone formation and maturation in the histological analysis, and a higher expression of BSP and osteocalcin in IHC. The optimal concentration of rhBMP-2 to the bone substitute material has not been reported. In vivo, 0.03 g of demineralized dentin matrix has been used with 5.0 µg of rhBMP-2, and it showed favorable new bone formation [40]. Hyaluronic acid hydrogel containing 30 µg of rhBMP-2 more effectively induces new bone formation and angiogenesis in the subperiosteal cranial rat model than that containing 1 µg of rhBMP-2 [19,41]. In this study, we used 0.08 g of ABB with 5 and 50 µg of rhBMP-2 and it effectively induced new bone formation. In particular, the group that had 50 µg of rhBMP-2 showed a greater amount and better quality of new bone than that of other groups.

From our animal study, we could evaluate the bone regeneration ability of rhBMP-2 in subperiosteal bone grafts by μ-CT and histological analysis. In previous clinical applications, the new bone formation in subperiosteal bone grafts has been evaluated just by postoperative radiograph [17,22]. A limitation of our in vivo study is that we used a rat calvarial model rather than a human clinical study. However, using this rat calvarial model, we could more exactly evaluate the amount of new bone formation by μ-CT and histological analysis and show the effect of rhBMP-2 on new bone regeneration in subperiosteal bone grafts. 

## 5. Conclusions

The combined use of the bone substitute and rhBMP-2 in the subperiosteal space has not been studied previously. In our study, we could confirm that rhBMP-2 contributes to new bone regeneration in the subperiosteal bone graft with ABB. The application of rhBMP-2 is an easy and simple procedure during subperiosteal bone grafts, and it effectively induces new bone formation around the ABB and the subperiosteal space. However, rhBMP-2 showed favorable osteoinductive ability in the subperiosteal bone graft, and further study would be required to find the best bone substitute scaffold and optimal concentration prior to clinical application.

In conclusion, we could confirm that rhBMP-2 can effectively and easily use the ABB substitute in the subperiosteal bone graft. In addition, this technique and clinical application of rhBMP-2 can be used effectively for new bone regeneration.

## Figures and Tables

**Figure 1 materials-12-01613-f001:**
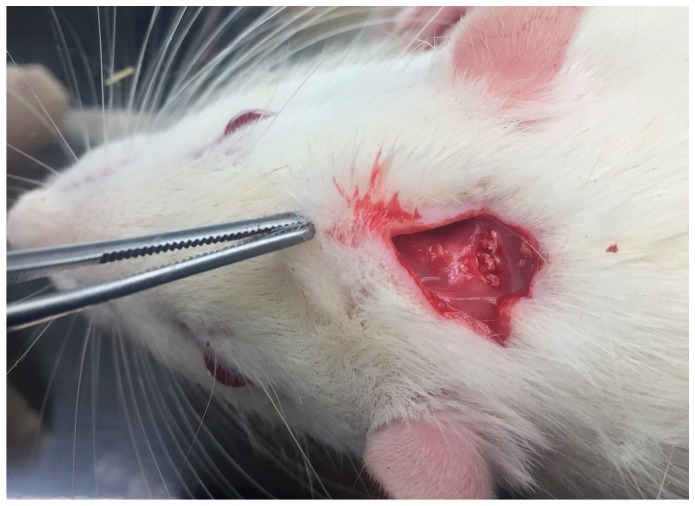
Surgical procedure of the subperiosteal bone graft. A horizontal step incision was made on the posterior portion of calvarial bone, and subperiosteal dissection was performed. A subperiosteal pocket was made on the parietal bone of calvarium, and the anorganic bovine bone (ABB) alone and the ABB with 5 and 50 µg of recombinant human bone morphogenetic protein-2 (rhBMP-2) were grafted in the pocket.

**Figure 2 materials-12-01613-f002:**
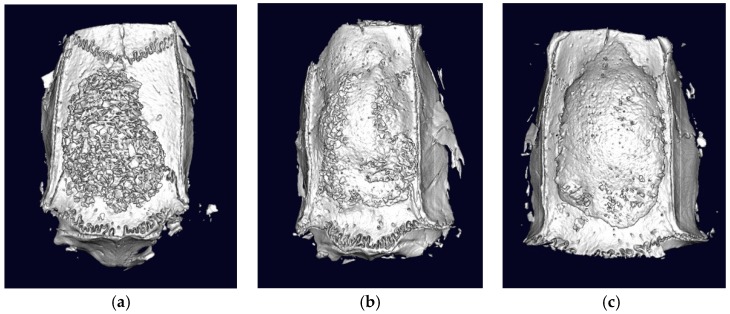
The 3D reconstruction images of microcomputed tomography (μ-CT) in the (**a**) control, (**b**) ABB/low bone morphogenetic protein (BMP), and (**c**) ABB/high BMP group at 6 weeks after operation.

**Figure 3 materials-12-01613-f003:**
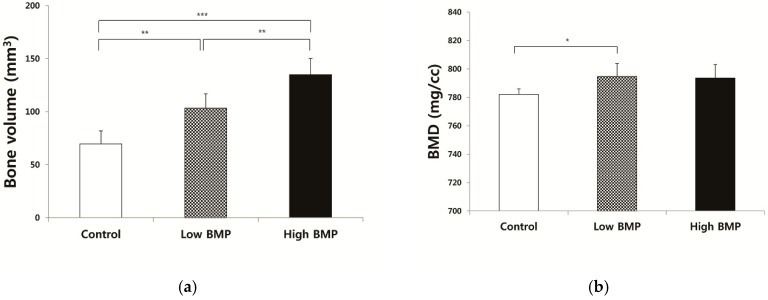
μ-CT analysis at 6 weeks after operation. (**a**) Bone volume, (**b**) bone mineral density, (**c**) trabecular thickness, and (**d**) trabecular spacing of the control (ABB: anorganic bovine bone), low BMP (ABB + 5 µg of rhBMP-2), and high BMP (ABB + 50 µg of rhBMP-2) group (** *p* < 0.05, *** *p* < 0.001).

**Figure 4 materials-12-01613-f004:**
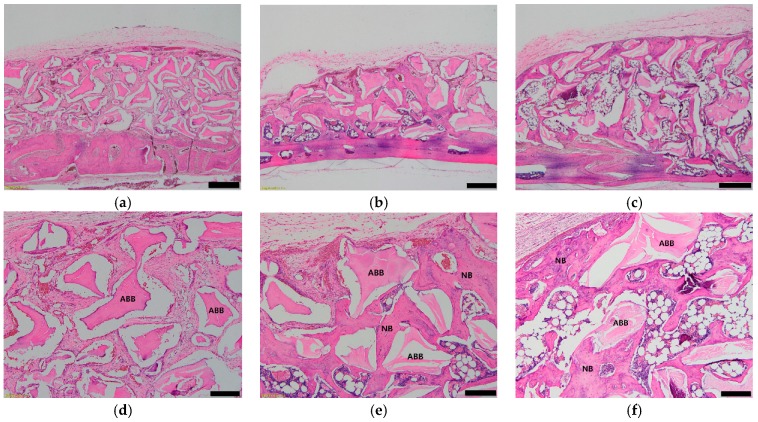
Histological images (hematoxylin and eosin staining) at 6 weeks after operation. (**a**,**d**) Control group, (**b**,**e**) ABB/low BMP, and (**c**,**f**) ABB/high BMP. (**d**–**f**) show high magnification of (**a**–**c**), respectively. (**a**–**c**): original magnification 40×, bar = 500 µm. (**d**–**f**): original magnification 100×, bar = 200 µm. ABB: anorganic bovine bone, NB: new bone.

**Figure 5 materials-12-01613-f005:**
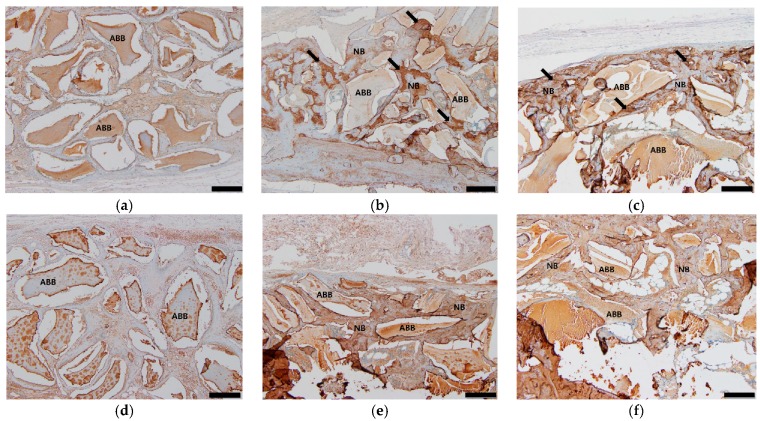
Immunohistochemical staining of bone sialoprotein (BSP; **a**–**c**) and osteocalcin (**d**–**f**) at 6 weeks after operation. (**a**,**d**) Control group, (**b**,**e**) ABB/low BMP, and (**c**,**f**) ABB/high BMP. A high expression of BSP and osteocalcin (black arrows) was observed in the new bone matrix of the ABB/low and high BMP groups (original magnification 100×, bar = 200 µm). ABB: anorganic bovine bone, NB: new bone.

**Table 1 materials-12-01613-t001:** Histomorphometric analysis of new bone formation area.

Group	Control	ABB/Low BMP	ABB/High BMP
Total new bone (%)	11.078 ± 3.10	31.03 ± 2.50	36.06 ± 6.92

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
