# Peer review of "Evaluation of the Bone Regeneration Effect of Recombinant Human Bone Morphogenic Protein-2 on Subperiosteal Bone Graft in the Rat Calvarial Model"

_materials, 2019, doi:10.3390/ma12101613_

Round 1
Reviewer 1 Report
The manuscript topic is actual and the paper has merit. It could be attractive, adequate and interesting for the journal readers. RhBmp2 experimental studies are very welcome and fit within the MATERIALS MDPI aim and scope.
However there are some point that authors should address in order to have a final more complete paper. Authors should underline the limitation of the value of the study, and the clinical and surgical implication of the presented study should be added. At this stage the paper seems to be directed to not clinical or surgeons readers. Please emphasize the clinical application of the study.
The limitation of an "animal study" should be underlined and need to be synthesized in a paragraph.
....animal studies will only become more valid predictors of human reactions to exposures and treatments if there is substantial improvement in both their scientific methods as well as in more systematic review of the animal literature as it evolves. Systematic reviews of animal research, if they are used to inform the design of clinical trials, particularly with respect to appropriate drug dose, timing and other crucial aspects of the drug regimen, will further improve the predictability of animal research in human clinical trials....
Introduction section should highlights the clinical rationale of this paper. Otherwise the study seems to be directed to just scientist or researcher and not to surgeons.
References are inadequate. Introduction section is poor. Some more references about the recent (2013-2019) CLINICAL reconstructive option just published to be added. Please add the following ones:
Cicciù M
J Craniofac Surg. 2017 May;28(3):592-593. doi: 10.1097/SCS.0000000000003595. Real Opportunity for the Present and a Forward Step for the Future of Bone Tissue Engineering.
At the same time discussion is poor.
In the discussion section authors should compare the results of the present study with others one presented and published in the literature. Other important bone substitutes material and clinical studies are the following, please add:
rhBMP-2 applied as support of distraction osteogenesis: A split-mouth histological study over nonhuman primates mandibles (Article)
Herford AS et al. International Journal of Clinical and Experimental Medicine
Volume 9, Issue 9, 30 September 2016, Pages 17187-17194
Author Response
-> Thanks for your kind advice. The clinical and surgical implication of this study was described in 4th paragraph of introduction and 6th paragraph of discussion. And the recommended drug dose was discussed in the 7th paragraph of discussion. According to your comments, we added one more paragraph some limitation of our animal study compared with clinical trials in last of the discussion.
-> Thanks for your kind advice. According to your comments, the clinical aspect and application of this study is important. In 4th paragraph of introduction, we described the clinical application of the subperiosteal bone graft on these days. The combine use of the rhBMP-2 in subperiosteal bone graft has not been reported, but the application of the rhBMP-2 is easy and simple procedure. The verifying the clinical efficacy of the rhBMP-2 in subperiosteal bone graft in vivo study would be helpful to the surgeon and it can be easily applicated in clinical field.
-> Thanks for your kind advice. We added this reference in our article.
Reviewer 2 Report
Review for Manuscript materials-507990-peer-review-v1
General Comments: Very nicely and clearly written; therefore extremely easy to read and review. I have just a few comments listed below in order since there are no line numbers.
More Specific Comments:
Title – None
Abstract
1) Change “in the rat model” to “in a rat model”
2) For “bone formation area”, anywhere in the manuscript discussing uCT it should be “volume” not “area”
3) Change “The subperiosteal bone graft” to “Subperiosteal bone grafts”
Introduction
1) Change “cysts” to “cyst”
2) Change “of alveolar bone” to “of the alveolar bone”
3) Change “For this reasons” to “For these reasons”
4) Change “The subperiosteal bone grafting” to “Subperiosteal bone grafting is”
5) Change “for a new bone” to “for new bone”
6) Change “of the bone regeneration marker” to “of bone regeneration markers”
Materials and Methods
1) Change “rhBMP-2 power” to “rhBMP-2 powder”
2) Change “was prepared” to “were prepared”
3) Under surgical procedure, drug names used should be their generic names
4) Change “and a histological analysis” to “and histological analysis”
5) For ROI, it should be VOI
6) Change “evaluate the expressions” to “evaluate the expression”
7) Great statistical analysis!
Results
1) Help the reader in the first paragraph by using a), b), and c) in the text to indicate the findings in Figure 2.
2) Saying there are significant differences between the groups is not necessary since in the next sentence you tell what the significant differences are.
3) With regard to the TbSp, instead of saying “significantly higher” it should just say “greater” since no statistical differences were found
4) Change “expression of the BSP” to “expression of BSP”
5) When discussing the osteoblasts and osteocytes, just say they “had high osteocalcin expression”
Discussion
1) Change “bone graft operation” to “bone graft operations”
2) Change “osteogenic cell” to “osteogenic cells”
3) Change “rhBMP-2 in the osteoblast differentiation” to “rhBMP-2 in osteoblast differentiation”
Conclusions – None
Figures, Tables, and Legends – None
Author Response
2) For “bone formation area”, anywhere in the manuscript discussing uCT it should be “volume” not “area”
-> Thanks for your kind advice. “bone formation area” in abstract was the result of the histomorphometric analysis that is calculated the newly formed bone area in histological image.
-> Thanks for your kind advice. We revised our manuscript according to your comments.
Reviewer 3 Report
Although this was not as thorough as the present study, the authors should probably cite:
Mokbel N, Naaman N, Nohra J, Badwawi N: Healing patterns of critical size bony defects in rats after grafting with bone substitutes soaked in recombinant human bone morphogenic protein-2: histological and histometric evaluation. Br J Oral Maxillofac Surg 51:545-549, 3013.
2. "The rhBMP-2 powder was diluted with saline, and the low (5ug) and high (50 ug) doses of rhBMP-2 was prepared." What volume of saline was used for what weight of rhBMP-2? What was then done to combine BMP with ABB?
3. "bone volume (BV)" This abbreviation is not needed sibce it is not subsequently used.
The size bars in Figures 4 & 5 would be better in black.
Author Response
Although this was not as thorough as the present study, the authors should probably cite:
Mokbel N, Naaman N, Nohra J, Badwawi N: Healing patterns of critical size bony defects in rats after grafting with bone substitutes soaked in recombinant human bone morphogenic protein-2: histological and histometric evaluation. Br. J. Oral Maxillofac. Surg. 51:545-549, 3013.
-> Thanks for your kind advice. We added this reference in our article.
2. "The rhBMP-2 powder was diluted with saline, and the low (5ug) and high (50 ug) doses of rhBMP-2 was prepared." What volume of saline was used for what weight of rhBMP-2? What was then done to combine BMP with ABB?
-> Thanks for your kind advice. We made 50 ug rhBMP-2/0.1 ml of saline and it diluted in 5 ug rhBMP-2/0.1 ml of saline. And 50 ug/0.1 ml and 5 ug/0.1 ml was injected on the ABB in the subperiosteal space. It was described in the 2.2. surgical procedure.
3. "bone volume (BV)" This abbreviation is not needed sibce it is not subsequently used.
-> Thanks for your kind advice. The abbreviation of BV was used 10th in our manuscript. I thought it would be better to describe bone volume as BV. But, if you do not want to say the BV, I will revise it.
The size bars in Figures 4 & 5 would be better in black
-> Thanks for your kind advice. I will revise it.
Round 2
Reviewer 1 Report
Authors made excellent job addressing all the reviewer comments